# 4-Azidocinnoline—Cinnoline-4-amine Pair as a New Fluorogenic and Fluorochromic Environment-Sensitive Probe

**DOI:** 10.3390/molecules26247460

**Published:** 2021-12-09

**Authors:** Natalia A. Danilkina, Ekaterina V. Andrievskaya, Anna V. Vasileva, Anna G. Lyapunova, Andrey M. Rumyantsev, Andrey A. Kuzmin, Elena A. Bessonova, Irina A. Balova

**Affiliations:** 1Institute of Chemistry, Saint Petersburg State University (SPbU), Universitetskaya nab. 7/9, 199034 Saint Petersburg, Russia; n.danilkina@spbu.ru (N.A.D.); vasilyeva1510@gmail.com (A.V.V.); fosfinomicin@mail.ru (A.G.L.); e.bessonova@spbu.ru (E.A.B.); 2Faculty of Chemistry, Belarusian State University, Leningradskaya Str. 14, 220006 Minsk, Belarus; kateandrievskaya@yandex.by; 3Department of Genetics and Biotechnology, Saint Petersburg State University (SPbU), Universitetskaya nab. 7/9, 199034 Saint Petersburg, Russia; rumyantsev-am@mail.ru; 4Institute of Cytology, Russian Academy of Sciences, St. Petersburg, Tikhoretsky Avenue 4, 194064 Saint Petersburg, Russia; a.kuzmin@incras.ru

**Keywords:** azides, amines, cinnolines, richer cyclization, fluorescence, fluorogenic probe, fluorochromic probe, environment-sensitive probe, AIE and ESPT fluorescence mechanisms

## Abstract

A new type of fluorogenic and fluorochromic probe based on the reduction of weakly fluorescent 4-azido-6-(4-cyanophenyl)cinnoline to the corresponding fluorescent cinnoline-4-amine was developed. We found that the fluorescence of 6-(4-cyanophenyl)cinnoline-4-amine is strongly affected by the nature of the solvent. The fluorogenic effect for the amine was detected in polar solvents with the strongest fluorescence increase in water. The environment-sensitive fluorogenic properties of cinnoline-4-amine in water were explained as a combination of two types of fluorescence mechanisms: aggregation-induced emission (AIE) and excited state intermolecular proton transfer (ESPT). The suitability of an azide–amine pair as a fluorogenic probe was tested using a HepG2 hepatic cancer cell line with detection by fluorescent microscopy, flow cytometry, and HPLC analysis of cells lysates. The results obtained confirm the possibility of the transformation of the azide to amine in cells and the potential applicability of the discovered fluorogenic and fluorochromic probe for different analytical and biological applications in aqueous medium.

## 1. Introduction

The chemistry of azides is a fast-growing research area. Organic azides are in great demand for the development of new synthetic methodologies [1,2,3], bioconjugation [4,5,6,7,8,9], photoaffinity labeling [10], synthesis of biologically active 1,2,3-triazole derivatives [11] and creation of different materials [12,13].

Fluorogenic probes are another important area for the application of organic azides [14,15,16]. This technique allows the conversion of non-fluorescent or weakly fluorescent materials to compounds with higher fluorescent ability. Fluorochromic compounds are able to respond to some changes in their structure or in the environment by a red-shift or a blue-shift of fluorescence [17].

In order to obtain fluorescent material from a non-fluorescent azide probe, the azido group can be modified in two ways: the conversion of azides into a 1,2,3-triazole derivatives by CuAAC [15,18,19,20,21,22,23] or SPAAC [23,24] (Figure 1A) or the reduction of azides to amines [23,24,25,26,27,28,29] (Figure 1B). 

The formation of fluorescent 1,2,3-triazoles from non-fluorescent azides is widely applied in bioimaging [15,18,19,20,21,22,30]. The reductive «azide to amine» transformation has been used for the detection of sulfide ions [25,26,27,28,31,32]. Moreover, it has been found that the azide group can be used as novel bioreductive functionality for the detection of intracellular hypoxia [33] and as a substrate for nitroreductases in microorganisms [29]. Recently, emissive in water quinoline-amines derived from corresponding azides as conjugates with peptides were discovered as probes for sensing the reactivity of protease K [34].

In most cases, fluorogenic organic azides possess an azide group attached either to an aromatic or to a heterocyclic ring. Thus, in (hetero)aromatic azides, the chemical transformations of the azido-group induce a reorganization of the whole conjugated electronic system that leads to the desired fluorogenic properties of the molecule. Several selected recent examples of fluorogenic azides of both types are represented in Figure 1A,B.

Recently, we have developed an efficient synthetic route towards 6-aryl-4-azidocinnolines based on the nucleophilic substitution of a bromine atom at the C4 position of 6-arylcinnolines with sodium azide [35]. The azidocinnolines possessed weak fluorescence (quantum yield (QY) < 1%); therefore, we believed that switching to the corresponding 4-(triazolyl)cinnolines would improve the fluorescent properties, providing a fluorogenic azide/triazole probe. However, 4-(triazolyl)cynnolines lacked any fluorescence, which was explained by fluorescence quenching through photoinduced electron transfer (PET) mechanism (Figure 1C, left).

In the current research, we tested the second type of fluorogenic transformation for 4-azidocinnoline—«azide to amine» reduction (Figure 1C, right) and found that the azide−amine pair is a new promising type of fluorogenic and fluorochromic environment-sensitive probe with the strongest increase of amine fluorescence in water.

## 2. Results

### 2.1. Synthesis of Cinnoline-4-amines

We chose a CN-substituted azide **5**/amine **6** pair as the object of the studies because of the direct conjugation of a CN group with N_3_/NH_2_ groups in azide **5** and amine **6**, respectively, which is crucial for the fluorescence through the intramolecular charge transfer (ICT) mechanism [17]. The synthesis of azide **5** was achieved by the synthetic protocol reported earlier [35] with a slight modification: the replacement of the chromatographic purification of 4-bromocinnoline **4** with recrystallization improved the yield of compound **4** from 62% up to 91% (Figure 1).

Two types of reducing conditions were tested: the Staudinger reduction and the use of sodium borohydride. The Staudinger reduction was found to be less appropriate, because it required separating the desired amine from triphenylphosphine oxide by-product and gave only a 55% yield of amine **6**. On the other hand, the reduction of azide **5** by NaBH_4_ in methanol afforded cinnoline-4-amine **6** in a quantitative yield without any by-products (Figure 1). Then, we tested the scope and limitations of this synthetic approach to cinnoline-4-amines. Thus, the same approach made it possible to successfully synthesize two additional derivatives of 6-halogen substituted cinnoline-4-amines **7a**,**b** in 85% (**7a**) and 74% (**7b**) yields at the last reduction step (Figure 1 and Scheme S1). These compounds can be considered as the basis for synthetic modification of cinnoline-4-amine scaffold through the halogen atom using various C−C cross-coupling reactions in the further design of 6-substituted cinnoline-4-amines with improved fluorescent properties.

### 2.2. Photophysical Properties of Cinnoline-Based Azide–Amine Pair

With the azide/amine **5/6** pair in hand, we investigated the photophysical properties of both compounds. First, the absorption spectra of the solutions of azide **5** and amine **6** in THF were measured (Figure 2, left). The obtained data revealed the obvious bathochromic shift of n→π* transition while switching from azide **5** (λ_max_ 317 nm) to amine **6** (λ_max_ 370 nm), which corresponds to higher donor ability of the NH_2_ group compared to the N_3_ group.

Then, we studied the absorption spectra of amine **6** in solvents with higher polarity, than THF as aprotic (DMSO, MeCN) as well as protic (*i*-PrOH, H_2_O) (Figure 2, right). Only small visible changes in UV spectra of the solutions of amine **6** in THF, MeCN and *i*-PrOH were observed. These data indicate the absence of crucial changes in dipole moments of ground and excited states of amine **6**. However, a small (10–15 nm) bathochromic shift was observed in the case of the solvents with higher polarity (DMSO and water) (Figure 2, right), which can be assumed as the attribute of intramolecular charge transfer (ICT).

Strong bathochromic shift of emission maximum was observed in the emission spectrum of THF solution of amine **6** (λ_amine_ ex/em = 375/450 nm) relative to azide **5** (λ_azide_ ex/em = 273/377 nm) (Figure 3, left) revealing the fluorochromic effect for the azide/amine pair. However, the fluorescence intensity of amine **6** was even lower than for the weakly fluorescent azide **5** [35].

To investigate solvatochromic effects on the fluorescent properties of amine **6** we used polar aprotic solvents (MeCN, DMSO) and protic solvents (*i*-PrOH and H_2_O) (Figure 3 right, Table 1). The switch from THF to MeCN did not change the fluorescence intensity; however, a small red-shift of the emission maximum was observed that could be an attribute of the ICT mechanism of amine’s **6** fluorescence [17]. A 3-fold increase in amine **6** fluorescence along with a bathochromic shift of fluorescence was detected for the solution of amine **6** in DMSO that also proves the ICT mechanism.

To our surprise, we observed almost the same increase in fluorescence intensity for the solution of amine **6** in *i*-PrOH compared to DMSO (Figure 3, right), along with the hypsochromic shift of fluorescence. The absolute QY for the solution of amine **6** in *i*-PrOH was higher (QY = 1.0%) than for the THF solution of amine **6** (QY = 0.2%) (Table 1). Thus, one can assume another mechanism of fluorescence in polar protic solvents due to the possible solvent–solute interactions through the forming intermolecular hydrogen bonding. [17,36] We suppose that the main fluorescence mechanism for amine **6** in isopropanol changes from ICT to excited state intermolecular proton transfer (ESPT) (Figure 2).

Thus, ESPT with the formation of quinoid-type emissive excited states was studied for different heterocyclic systems [37,38,39,40] and a competition between ICT and ESPT for heterocyclic D-π-A molecular architecture has been reported recently [41].

Finally, a huge increase in the fluorescence intensity (6−85 times) was detected for the solutions of amine **6** in water in the presence of 1% *v*/*v* of the appropriate solvent (DMSO, *i*-PrOH, THF) (Figure 4, right) with the strongest fluorescence amplification (85 times) for the aqueous solution prepared from the stock solution in THF (QY**^6^**_H2O/THF_ = 11.0%). It should be noted, that the aqueous solution of azide **5** lacked the fluorogenic properties (Figure 4, left) (QY**^5^**_H2O_ = 0.0%). The remarkable fluorogenic properties for the azide/amine pair in water, and the fact that the maximum of amine **6** emission in water (425 nm) is outside the background fluorescence of proteins and DNA, can be considered as a promising opportunity for practical application of the discovered new fluorogenic probe.

The revealed dramatic increase in the fluorescent properties of aqueous solutions of amine **6** forced us to seek an explanation. On the one hand, the observed increase in fluorescence is fully consistent with the ESPT fluorescence mechanism because water can also interact with the solute through the formation of hydrogen bonds [37]. On the other hand, aggregation-induced emission (AIE) should not be ruled out [42,43,44]. In this case, water is considered as an aggregating solvent, while DMSO, *i*-PrOH or THF are known to be non-aggregating solvents, and the aggregates formed are able to fluoresce as a result of a reduced possibility of nonradiative relaxation. Thus, different compounds with nitrile groups have been recently discovered as efficient AIE luminogens (or AIEgens) [45,46,47]. However, there is also the possibility for the combination of AIE with ESPT. The weaker fluorescence enhancement in the case of the aqueous solutions of amine **6** prepared from DMSO stock solution (5.9 times) compared to *i*-PrOH (21.5 times) and THF (84.9 times) stock solutions (Figure 4, right) can be viewed as the evidence for the cooperation of AIE and ESPT. In the former case, the residual polar aprotic non-aggregating solvent (DMSO) can interfere with the formation of aggregates through hydrogen bonding with water molecules, which leads to less fluorescence intensity compared with THF-water mixtures.

We carried out two additional experiments to elucidate the mechanism of the induced fluorescence. Thus, aqueous solutions of amine **6** prepared from stock solutions (DMSO and *i*-PrOH) were filtered through nylon (hydrophilic) or polytetrafluoroethylene (PTFE, lipophilic) membranes before the absorption and emission measurements. As a result, a significant decrease in both the absorption and emission intensities was observed in each case (Figure 5). The obtained data revealed that nano-sized aggregates are formed in both aqueous solutions. The aggregates can be removed by filtration through the nylon filters more efficiently, than by using a PTFE filter, which proves the hydrophilic nature of aggregates, i.e., they are formed from the amine **6** and water molecules. The lower decrease in fluorescence of filtered water solution of amine **6** prepared from DMSO proves the lower stability of aggregates due to the solvation with residual DMSO along with the decreased size of these aggregates compared to the aggregates formed in the presence of *i*-PrOH.

Then the emission spectra for the solutions of amine **6** in *i*-PrOH-H_2_O mixtures with different water fraction (f_w_) (in vol%) were measured. The obtained data illustrate the direct gradual dependence of fluorescence intensity on the water fraction (Figure 6, left). It is important to note, that typical AIE curves reveal faint emission at water fractions below a definite meaning, and progressively increasing emission as water fraction exceeds this value [42]. Therefore, the gradual increase in fluorescent intensity in the case of amine **6** (Figure 6, right) can be considered as the additional evidence for the cooperation between AIE and ESPT mechanisms. Moreover, the absence of fluorescence of amine **6** in solid state also proves that AIE is not the only reason for the induced fluorescence in water [42].

Therefore, when polar protic solvents were used for the detection of amine emission, both fluorochomic and fluorogenic properties for the azide/amine pair were discovered. The observed sensitivity of amine fluorescence to the environment opens the way for the use of the discovered azide/amine pair as a sensor system in aqueous media and in biological systems.

### 2.3. In Vitro Study of the Behavior of the 4-Azodicinnoline/Cinnoline-4-amine Pair 

Taking into account the growing interest in a wide range of biological applications of fluorogenic azide/amine pairs [29,31,32,33], we were interested in whether the discovered fluorogenic probe based on the 4-azidocinnoline/cinnolin-4-amine pair retains its fluorogenic properties in living cells.

To test the cell penetration ability of azide **5** and amine **6**, as well as the possibility of the in vivo reduction of azide **5** to fluorescent amine **6**, we selected the HepG2 liver cancer cell line, as it was used to detect intracellular hypoxia by measuring the fluorescence of an amine derived from the corresponding azide-containing coumarin fluorogenic and fluorochromic probes [33].

Two separate cell probes were treated with amine **6** and azide **5** (c = 10 μM), respectively, and incubated for 4 h at 37 °C. Then, both probes were analyzed by fluorescent microscopy and flow cytometry. The obtained data revealed that amine **6** can enter HepG2 cells, and it accumulates in different cell organelles that can be detected by fluorescent microscopy using standard 4′,6-diamidino-2-phenylindole (DAPI) light cube (357/44 nm excitation; 447/60 nm emission) (Figure 7A) and by flow cytometry (Figure 7B). Fluorescence microscopic images of cell monolayers (Figure 7A) show fluorescent cells with non-fluorescent cell nuclei, suggesting that amine **6** enters cells but does not accumulate in the cell membrane and cell nuclei.

Encouraging results were obtained in experiments with azide **5**: the fluorescence in the same wavelength region, but with less intensity, was detected that illustrates the «azide to amine» conversion upon cells incubation and the possibility of the detection of the amine **6** derived from the azide **5** in cells. The quantitative comparison of the intensities of cells fluorescence was achieved by fluorescence-activated cell sorting (FACS) analysis of both probes, which yielded higher fluorescence of cells treated with amine **6** (Figure 7B). 

To prove that the observed fluorescence of the probe pre-incubated with azide **5** is the evidence of the «azide to amine» conversion in vitro, the probe of was washed from extracellular organic components and subjected to cell lysis by addition of MeCN. The lysate obtained was analyzed by HPLC analysis. In the cell lysate, we found a higher concentration of amine **6** than the starting azide **5** (3.60 ± 0.06 and 2.92 ± 0.09 μM, respectively). Calibration data and the typical chromatograms of the standards solutions and the samples of cell lysate are represented in SI (Appendix A). Hence, it is obvious, that in vitro «azide to amine» conversion occurred. 

## 3. Discussion

A new fluorogenic and fluorochromic azide/amine pair based on a cinnoline core was discovered. 4-Azido-6-(4-cyanophenyl)cinnoline was converted to corresponding cinnoline-4-amine by the reduction with sodium borohydride in a quantitative yield. This amine has improved fluorescent properties compared to the starting azide: a red shift of the emission (fluorochromic effect) was observed in solvents with differing polarities, while an increase in fluorescence intensity (fluorogenic effect) was found in polar solvents, with the strongest one in water.

The measurements of fluorescence of water solutions of amine **6** prepared from the stock solutions in solvents with different nature before and after the filtration through the membrane filters with different polarity as well as the fluorescence measurements for the solutions with different water fraction prove the cooperative assistance of both AIE and ESPT as the fluorescence mechanism for the aqueous solutions of amine **6**. Therefore, the formation of associates of amine **6** molecules through the hydrogen bonding with water molecules can be assumed (Figure 3, A-form).

Recently, Wu et al. explained the ESPT mechanism from the point of view of the antiaromaticity of the excited states for [4n + 2] π-electron delocalized systems (A-forms) (Baird’s rule) [37]. To relieve the antiaromaticity, the conversion of such systems (A-forms) into quinoid-type forms (Q-forms) occurs through either inter- or intramolecular proton transfer in the exited state. We suppose that in the case of associates of amine **6** with water molecules (Figure 3, A-form), quinoid-type aggregates of amine **6** with water (Figure 3, Q-form) can form in the exited state, which leads to the increased ratio of emitting to non-emitting transitions.

The biological trials revealed that fluorescence of amine **6** in aqueous medium also appears in complex systems, i.e., in HepG2 cell culture. Moreover, amine **6** can penetrate cell membranes and accumulate within cells except cell nuclei. The QY of amine **6** in water (11%) is sufficient for the detection of cell fluorescence.

The conversion of azide **5** to amine **6** in a culture medium detected by fluorescent microscopy and FACS was proved by HPLC analysis of cell lysate. Considering that aromatic azides are known to be substrates for different enzymes (nitroreductases in microorganisms, [29] bactosomal CYP450 enzymes (CYP 1A2, 2D6, 3A4, 2C9, 2C19)), which are able to reduce aromatic azides to amines, and since aromatic azides can undergo H_2_S-mediated reduction under various conditions in in vitro and in vivo settings [48], the discovered azide−amine pair with a cinnoline core may find a range of biological applications, including intracellular reductase activity assays.

## 4. Materials and Methods

Solvents and reagents used for reactions were purchased from commercial suppliers. Catalyst Pd(PPh_3_)_4_ was purchased from Sigma-Aldrich (München, Germany). Chemicals were used without further purification. 4-Chloro-2-iodoaniline [49] and ethynytltriazene **1a** [35,50] were synthesized by known procedures without any modifications. Evaporation of solvents and concentration of reaction mixtures were performed in vacuum at 35 °C on a rotary evaporator. Thin-layer chromatography (TLC) (Merck, Darmstadt, Germany) was carried out on silica gel plates (Silica gel 60, F254, Merck) with detection by UV or staining with a basic aqueous solution of KMnO_4_. Melting points (mp) determined are uncorrected. ^1^H and ^13^C NMR spectra were recorded at 400 and 100 MHz, respectively, at 25 °C in acetone-*d*_6_ or CDCl_3_ or DMSO-*d*_6_ without the internal standard using a Bruker 400 MHz Advance spectrometer (Bruker, Billerica, MA, USA). The ^1^H NMR data are reported as chemical shifts (δ), multiplicity (s, singlet; d, doublet; t, triplet; q, quartet; m, multiplet; br, broad), coupling constants (*J*, given in Hz) and number of protons. The ^13^C NMR data are reported as the chemical shifts (δ). Chemical shifts for ^1^H and ^13^C are reported as δ values (ppm) and referenced to residual solvent (δ = 2.05 ppm for ^1^H; δ = 29.84 for ^13^C—for spectra in acetone-*d_6_*, δ = 7.26 ppm for ^1^H; δ =77.16 ppm for ^13^C—for spectra in CDCl_3_ and δ = 2.50 ppm for ^1^H; δ =39.52 ppm for ^13^C—for spectra in DMSO-*d_6_*). High-resolution mass spectra (HRMS) were determined using electrospray ionization (ESI) in the mode of positive ion registration with a TOF mass analyzer (Bruker Maxis Q-TOF). UV–vis spectra for solutions of all compounds in appropriate solvents were recorded on a Shimadzu UV-1800 spectrophotometer at 25 °C. Fluorescence spectra for the same solutions were recorded on a Horiba Scientific FluoroMax-4 spectrofluorometer at 25 °C.

### 4.1. Synthetic Procedures

**(E)-4′-(3-ethyl-3-phenyltriaz-1-en-1-yl)-3′-(hept-1-yn-1-yl)-[1,1′-biphenyl]-4-carbonitrile (3):** Triazene **1a** [35] (530 mg, 1.26 mmol, 1.00 equiv), 4-cyanophenylboronic acid **2** (293 mg, 1.89 mmol, 1.50 equiv), K_3_PO_4_ (565 mg, 2.52 mmol, 2.00 equiv), and Pd(PPh_3_)_4_ (76.9 mg, 0.0665 mmol, 5.00 mol%) were placed in a vial. The vial was sealed, and the mixture was degassed using a freeze-pump-thaw technique over three degassing cycles. 1,4-Dioxane (12.6 mL, c = 0.100 M) was added to the vial with a syringe, and the vial was placed in a preheated oil bath (80 °C). The reaction mixture was stirred at 80 °C for 3 h (TLC control). After completion of the reaction, the reaction mixture was cooled, filtered through a pad of silica gel, the sorbent was washed with ethyl acetate and the resulting solution was concentrated under reduced pressure to give the crude product, which was purified by column chromatography on silica gel using hexane/EtOAc/Et_3_N (30:1:0.01) as the eluent to give **3** (335 mg, 60%) as a yellow oil. ^1^H NMR (acetone-*d*_6_): = δ 7.96–7.83 (m, 4H), 7.79 (d, *J* = 2.0 Hz, 1H), 7.76–7.63 (m, 2H), 7.62–7.56 (m, 2H), 7.50–7.38 (m, 2H), 7.21–7.15 (m, 1H), 4.45 (q, *J* = 7.1 Hz, 2H), 2.49 (t, *J* = 7.0 Hz, 2H), 1.66–1.59 (m, 2H), 1.55–1.41 (m, 2H), 1.40–1.31 (m, 5H), 0.89 (t, *J* = 7.3 Hz, 3H).

**General Procedure for the Richter-type cyclization:** To a solution of triazene (1 equiv) in acetone (c = 0.1 M), HBr (48% aqueous solution, 20 equiv) was quickly added dropwise, maintaining the temperature at 20 °C by cooling the reaction mixture with a water bath. The resulting mixture was stirred for 10 min. Then, the reaction mixture was diluted with an aqueous solution of triethylamine (21 equiv). The product was isolated either by filtration followed by washing with water or by extraction. In the case of extraction, the mixture was extracted with ethyl acetate three times. The combined organic layers were washed with water, then brine and dried over anhydrous Na_2_SO_4_. The solvent was removed under reduced pressure, and the crude product was purified either by column chromatography or by the recrystallization.

**4-(4-Bromo-3-pentylcinnolin-6-yl)benzonitrile (4):** Cinnoline **4** was synthesized in accordance with the general procedure for the Richter-type cyclization from triazene **3** (335 mg, 0.880 mmol, 1.00 equiv). The crude product—a yellow solid (279 mg, 92%) was filtered off, washed with water, dried and used in the next step without purification. The sample for the m. p. determination was recrystallized from acetonitrile. m.p. 135−137 °C. The spectral data correspond to the data reported earlier [35]. ^1^H NMR (400 MHz, acetone-*d*_6_) δ = 8.60 (d, *J* = 8.8 Hz, 1H), 8.44–8.40 (m, 1H), 8.32 (dd, *J* = 8.8 Hz, 1.9 Hz, 1H), 8.15–8.10 (m, 2H), 8.02–7.97 (m, 2H), 3.45–3.38 (m, 2H), 1.98–1.87 (m, 2H), 1.54–1.34 (m, 4H), 0.93 (t, *J* = 7.0 Hz, 3H).

**General Procure for the Nucleophilic Substitution:** Sodium azide (5.00 equiv) was added to a solution of 4-bromocinnoline (1.00 equiv) in absolute DMF (c = 0.1 M). The mixture was degassed and stirred under Ar at 50 °C for 24 h (TLC control). Upon completion of the reaction, the reaction mixture was poured into water and extracted with ethyl acetate three times. The combined organic layers were washed three times with water and two times with brine, dried over anhydrous Na_2_SO_4_. The solvent was removed in vacuum to yield the crude product, which was purified by column chromatography on silica gel.

**4-(4-Azido-3-pentylcinnolin-6-yl)benzonitrile (5):** Cinnoline **5** was synthesized in accordance with the general procedure for the nucleophilic substitution from 4-bromocinnoline **4** (260 mg, 0.684 mmol, 1.00 equiv) and NaN_3_ (222 mg, 3.42 mmol, 5.00 equiv). Purification of the crude product by column chromatography using hexane/EtOAc (5:1) as the eluent gave **5** (183 mg, 78%). Decomp. at 90 °C. The spectral data correspond to the data reported earlier [35]. ^1^H NMR (400 MHz, acetone-*d*_6_) δ = 8.57–8.47 (m, 2H), 8.27 (dd, *J* = 8.9 Hz, 2.0 Hz, 1H), 8.15–8.04 (m, 2H), 8.02–7.92 (m, 2H), 3.46–3.22 (m, 2H), 2.00–1.88 (m, 2H), 1.56–1.36 (m, 4H), 0.93 (t, *J* = 7.1 Hz, 3H).

**General Procure for the Azide to Amine reduction:** NaBH_4_ (1.50−3.00 equiv) was added to a well stirred solution of azidocinnoline (1.00 equiv) in MeOH cooled to 0 °C. The resulting mixture was allowed to warm to room temperature and stirred at this temperature for 1 h (TLC control). After completion of the reaction, MeOH was removed in vacuum, and the residue was dispersed in a small amount of ethyl acetate. The resulting mixture was filtered through the pad of silica gel. The EtOAc fractions were discarded, and the pure amine was washed from silica gel by MeOH. The evaporation of the solution of amine to dryness in vacuum gave pure amine without need of further purification. The sample for the photophysical and biological studies was recrystallized.


**4-(4-Amino-3-pentylcinnolin-6-yl)benzonitrile (6):**


**(A)** Amine **6** was obtained in accordance with the general procedure for the reduction from azide **5** (40.0 mg, 0.117 mmol, 1.00 equiv) and NaBH_4_ (6.65 mg, 0.175 mmol, 1.50 equiv) in methanol (1.17 mL). The crude product (38.1 mg, 97%) was pure and did not require additional purification. The sample for the photophysical and biological studies was recrystallized from acetone. m.p. 205−207 °C. ^1^H NMR (400 MHz, DMSO-*d*_6_): δ = 8.71 (d, *J* = 1.4 Hz, 1H), 8.13−8.04 (m, 4H), 8.02−7.96 (d, *J* = 8.5 Hz, 2H), 7.05 (s, 2H), 3.07 − 2.93 (m, 2H), 1.81−1.66 (m, 2H), 1.45−1.26 (m, 4H), 0.87 (t, *J* = 7.0 Hz, 3H). ^13^C NMR (101 MHz, DMSO-*d*_6_): δ = 147.6, 143.5, 142.3, 139.2, 135.9, 132.8, 129.0, 127.8 (two signals of a cinnoline cycle are overlapped with each other), 120.5, 118.8, 114.5, 110.4, 31.2, 31.0, 27.1, 22.1, 14.0. HRMS (ESI): *m*/*z* calcd for C_20_H_19_N_4_+H^+^: 317.1761 [M + H]^+^; found: 317.1750.

**(B)** To a well-stirred solution of azidocinnoline (40.0 mg, 0.117 mmol) in MeOH (3.00 mL), triphenylphosphine (46.0 mg, 0.175 mmol, 1.50 equiv) was added. The reaction mixture was stirred at room temperature for 30 min and then treated with an aqueous solution of acetic acid (AcOH (176 mg, 0.168 mL, 2.93 mmol, 25.0 equiv) and H_2_O (0.252 mL)). After completion of the reaction (TLC control), the reaction mixture was poured into 10% aqueous solution of NaHCO_3_ (20.0 mL) and extracted with ethyl acetate (3 × 10.0 mL). The combined organic layers were washed with brine (30.0 mL) and dried over anhydrous Na_2_SO_4_. The solvent was removed under reduced pressure, and the crude product was purified by the recrystallization from acetone that gave amine **6** (21.6 mg, 55%).

### 4.2. Biological Studies

**Cell cultivation and treatment with azide 5 and amine 6.** The HepG2 hepatic cancer cell line was obtained from the shared research facility “Vertebrate cell culture collection” supported by the Ministry of Science and Higher Education of the Russian Federation (Agreement No. 075-15-2021-683). Cells were routinely cultured in DMEM (Biolot) medium containing 10% FBS (Sigma), penicillin (100 U/mL) and streptomycin (100 μg/mL) (Gibco) under standard conditions (37 °C, 21% O_2_ and 5% CO_2_). For azide **5** and amine **6** treatment, cells were seeded in 6-well plates and cultured in 2 mL of the growth media to obtain a concentration of 5 × 10^5^ cells per well. Then, culture medium was aspirated and replaced with 2 mL of growth medium containing 0.5% DMSO and 10 μM of azide **5** or amine **6**. After medium replacement, cells were incubated for 4 h at standard conditions.

**Fluorescence microscopy.** After incubation, media containing substances was aspirated. Cells were washed with culture medium and imaged using the EVOS FL Auto Imaging System (Thermo Fischer Scientific) equipped with the DAPI Light cube (357/44 nm excitation; 447/60 nm emission).

**Flow Cytometry.** For flow cytometry, the probes after the fluorescence microscopy were washed with PBS and treated with 0.25% Trypsin-EDTA (Gibco) for 10 min at 37 °C. Then, cells were harvested and resuspended in 1 mL of culture medium. Downstream analysis was performed using CytoFLEX Flow Cytometer (Beckman Coulter) with the following fluorescence parameters: 375 nm excitation and 450/45 emission. 

**Cell Lysis.** Cells were washed with PBS and treated with 0.25% Trypsin-EDTA (Gibco) for 10 min at 37 °C. Then, cells were harvested and lysed in 50 µL of MeCN. Cell debris was removed by centrifugation (12,000× *g*), and the supernatant was used for HPLC analysis.

### 4.3. The HPLC Analysis of Cells Lysate

HPLC analysis was performed using a liquid chromatograph LC-20 (Prominance, Shimadzu, Japan) with photodiode array detector (DAD) and fluorescence detector (FL) on an analytical Phenomenex Luna C18(2) column (5 µm, 150 × 2.1 mm). The analysis results were processed using the LCsolution software. Chromatographic conditions for the separation of amine **6** and azide **5**: water with 0.1% HCOOH/CH_3_CN; gradient elution mode from 30 to 70% of CH_3_CN for 10 min; injected sample volume−2 μL; UV detection: 275 nm, FL detection: λ_ex_/λ_em_ for amine **6**—380/456 nm, λ_ex_/λ_em_ for azide **5**—273/373 nm.

Calibration curve for each analyte was constructed by plotting peak area of the analyte versus the analytes amount and the fitting was done by linear regression as presented in Appendix A. The standard chromatogram of amine **6** and azide **5** and their mixture are illustrated in Appendix A.

### 4.4. The Measurements of Absolute Fluorescence Quantum Yields of Azide **5** and Amine **6**

The absolute fluorescence quantum yield (QY) was measured on spectrometer Horiba Fluorolog-3 equipped with integrating sphere. The xenon lamp coupled to a double monochromator was used as excitation light source. The sample solutions of azide **5** or amine **6** (1 cm quartz cuvette cell with molecular solution in THF) or blank (pure THF) were directly illuminated in the center of the integrating sphere. The optical density of the investigated sample solutions did not exceed 0.1 at the luminescence excitation wavelength. The stock solutions of either azide **5** or amine **6** in THF (c = 1 × 10^−3^ mol/L) were used to prepare the working solutions of azide **5** (5 × 10^−6^ mol/L) and amine **6** (1.4 × 10^−6^ mol/L) in the appropriate solvents (Appendix A). Under the same conditions (e.g., excitation wavelength, spectral resolution, temperature), the luminescence spectrum of the sample *E_c_*, the luminescence spectrum of the blank *E_a_*, the Rayleigh scattering spectrum of the sample *L_c_*, and the Rayleigh scattering spectrum of the solvent *L_a_* were measured. The absolute fluorescence quantum yield was determined according to the formula:QY = (*Ec* − *Ea*)/(*La* − *Lc*)

## 5. Conclusions

In summary, a new fluorogenic and fluorochromic azide/amine pair based on a cinnoline core was developed. 4-Azido-6-(4-cyanophenyl)cinnoline was converted to corresponding cinnoline-4-amine by reduction with sodium borohydride in a quantitative yield. This amine has improved fluorescent properties compared to the starting azide: a red shift of the emission (fluorochromic effect) was observed in solvents with differing polarities, while the highest increase in fluorescence intensity (fluorogenic effect) was detected in protic solvents. The fluorogenic properties of cinnoline-4-amine in water are determined by the cooperation of two mechanisms: excited state intermolecular proton transfer (ESPT) and aggregation-induced emission (AIE). As a result, cinnolin-4-amines can be viewed as a novel environmentally-sensitive fluorescent probe, while the azide-to-amine transformation may hold promise for the development of various sensory systems, including a biosensor. The developed synthetic approach opens the way for the search and design of new cinniolin-4-amines with optimal photophysical properties for the specific biological applications. These studies are ongoing.

## Data Availability

Data is contained within the article and Appendix A.

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
