# Peer review of "4-Azidocinnoline—Cinnoline-4-amine Pair as a New Fluorogenic and Fluorochromic Environment-Sensitive Probe"

_molecules, 2021, doi:10.3390/molecules26247460_

Round 1

Reviewer 1 Report

The manuscript “4-Azidocinnoline / Cinnoline-4-amine pair as a New Fluorogenic / Fluorochromic Environment-Sensitive Probe” by N. Danilkina and co-authors describes synthesis and photophysical properties of 6-substituted cinnoline-4-amine derivatives via corresponding azide compounds. It reported that the amine 6 exhibited strong fluorescence emission in polar solvents whereas the azide 5 showed weak fluorescence. The AIE property of amine 6 in water solution is interesting. The other compounds synthesized in this study are expected to exhibit curious fluorescence properties. I recommend acceptance for publication in Molecules, however I have one comment concerning fluorescence mechanism. The authors describe that the weak fluorescence of amine 6 in THF and MeCN is explained by ICT model. On the other hand, I wonder that there is another quenching mechanism by tautomerization of amine 6 as shown in Scheme 4. As such, the fluorescence in DMSO and iPrOH may be recovered by solvation. Finally, please check following minor points.

Minor comments:

P7-L241−243; “(Fig. S3)” should be added in the sentence.

P11-L366; 8.02 − 7.96? (d, J = 8.5 Hz, 1H 2H?)

Author Response

REV 1 The manuscript “4-Azidocinnoline / Cinnoline-4-amine pair as a New Fluorogenic / Fluorochromic Environment-Sensitive Probe” by N. Danilkina and co-authors describes synthesis and photophysical properties of 6-substituted cinnoline-4-amine derivatives via corresponding azide compounds. It reported that the amine 6 exhibited strong fluorescence emission in polar solvents whereas the azide 5 showed weak fluorescence. The AIE property of amine 6 in water solution is interesting. The other compounds synthesized in this study are expected to exhibit curious fluorescence properties. I recommend acceptance for publication in Molecules, however I have one comment concerning fluorescence mechanism. The authors describe that the weak fluorescence of amine 6 in THF and MeCN is explained by ICT model. On the other hand, I wonder if there is another quenching mechanism by tautomerization of amine 6 as shown in Scheme 4. As such, the fluorescence in DMSO and iPrOH may be recovered by solvation. Finally, please check the following minor points.

Tree different types of fluorescence are consumed. The first one is intramolecular charge transfer (ICT), which is observed in aprotic solvents (THF, MeCN, DMSO). The second one is Excited State Intermolecular proton transfer (ESPT) which is observed in isopropanol (Scheme 2 in the revised version)  – only aromatic form (A-form) is shown. Proton transfer occurs in the excited state with the formation of a corresponding Q-form. (See Scheme 3 in the revised version). And the third one is a combination of Aggregation-Induced Emission (AIE) and ESPT which is observed in water after mixing of stock solution of amine 6 in non-aggregating solvent (THF or DMSO or i-PrOH) with the aggregating solvent (water). Therefore in scheme 4 there is not a fluorescence quenching process, but it is the induction of fluorescence. We suppose that the aggregation occurs through the association with water molecules that raises the fluorescence by a proton transfer in the excited state. The proton transfer leads to the formation of highly emissive aggregates with a quinoid-type (not antiaromtic) structure. For the details of processes please see Ref. 37  (revised version) Wu, C.-H.; Karas, L.J.; Ottosson, H.; Wu, J.I.C. Excited-state proton transfer relieves antiaromaticity in molecules. Proc. Natl. Acad. Sci. 2019, 116, 20303–20308.

Minor comments:

P7-L241−243; “(Fig. S3)” should be added in the sentence.

References to all figures with chromatograms were added in the text

P11-L366; 8.02 − 7.96? (d, J = 8.5 Hz, 1H 2H?)

2H, done

Reviewer 2 Report

  1. In introduction part, the sentence in line 29-35 and line 41-43 are redundant and there are not the advantages of using this probe, please try to convince more about the importance of the probe usage.
  2. The information of compound 2 is not found in condition (a) of scheme 1.
  3. Which compound is referred to “0.1 M” in condition in scheme 1 (b)?
  4. What is the ratio between Acetic acid/H2O in condition in scheme1 (d)?
  5. The writing style of scheme 1 and 2 are inconsistent, like (a) and a- , use like in scheme 1 is more clear.
  6. Please descript the meaning of n.a. below Table 1.
  7. The descripted figures are missing in many places in the text, e.g. no (Figure 2b) in line 120. Please fill in all figures that you have mentioned in the text.
  8. Figure S3 has not been mentioned in the text.
  9. No discussion about the biological application (2.3) is found.
  10. The experimental data of in vitro experiment was not found in the manuscript and the supporting information.
  11. In line 491, could be Table S1 and Fig. S1-2?
  12. Please mention the retention time of amine 6 in Fig. S1 and retention time of azide 5 in Fig. S2 as well as explain about the inserted table in Fig. S2.

Author Response

REV 2 In introduction part, the sentence in line 29-35 and line 41-43 are redundant and there are not the advantages of using this probe, please try to convince more about the importance of the probe usage.

Lines 29-43 were edited and rewritten. New references (1-3,11-13) were added to the Introduction. New Ref about the usage of azide-amine probe were added (29,31,32,34)

The information of compound 2 is not found in condition (a) of scheme 1.

Structural formula of 2 is represented above the arrow. There is no need to mention it in conditions a.

Which compound is referred to “0.1 M” in condition in scheme 1 (b)?

It is concentration of triazene 3. The concentration was removed because all details are provided in the Experimental Section lines 317-318 in the original version

What is the ratio between Acetic acid/H2O in condition in scheme1 (d)?

All details are given in the Experimental Section. See lines 373-374 in the original version

«solution of acetic acid (AcOH (176 mg, 0.168 mL, 2.93 mmol, 25.0 equiv) and H2O (0.252 mL))».

The writing style of scheme 1 and 2 are inconsistent, like (a) and a- , use like in scheme 1 is more clear.

Done, Scheme 2 was moved to SI

Please descript the meaning of n.a. below Table 1.

done

The descripted figures are missing in many places in the text, e.g. no (Figure 2b) in line 120. Please fill in all figures that you have mentioned in the text.

done

No discussion about the biological application (2.3) is found.

Corrected

The experimental data of in vitro experiment was not found in the manuscript and the supporting information.

The description is given in lines 458-480 (please see the first version for lines numbering)

In line 491, could be Table S1 and Fig. S1-2?

done

Please mention the retention time of amine 6 in Fig. S1 and retention time of azide 5 in Fig. S2 as well as explain about the inserted table in Fig. S2.

We mentioned the retention time for all chromatograms with standards and mixtures (amine 6 + azide 5)

The inserted table illustrates the Integral Intensity of peaks observed in the Chromatogram, and therefore the HPLC purity of azide 5 which is 97.7%. It was removed in the current version

Reviewer 3 Report

A nice study of a new AIE/responsive probe that can in theory undergo insitu reduction to light up. The description is good and the mechanistic exploration interesting. My only queries are (i) whether the authors can demonstrate that insitu reduction ca be controlled by some external stimulus on the cell, or that it occurs more readily in some parts of the cell than others (i.e. is the probe useful) and (ii) whether there is an environmental response of the probe fluorescence across the cell organelles (e.g. cytoplasm vs cell membrane vs vesicle)?   

Author Response

REV 3 A nice study of a new AIE/responsive probe that can in theory undergo insitu reduction to light up. The description is good and the mechanistic exploration interesting. My only queries are (i) whether the authors can demonstrate that insitu reduction ca be controlled by some external stimulus on the cell, or that it occurs more readily in some parts of the cell than others (i.e. is the probe useful)

and (ii) whether there is an environmental response of the probe fluorescence across the cell organelles (e.g. cytoplasm vs cell membrane vs vesicle)?

Experiments using confocal microscopy and fluorescence lifetime imaging microscopy, which will allow the analysis of the environmental response of the probe fluorescence across the cell, are planned as the next part of a study on the biological properties of substances.

Reviewer 4 Report

The manuscript by Natalia A. Danilkina describes the synthesis, characterization and application of a new cinnoline based fluorescent probe that is sensitive to the environment. Overall I found the manuscript to be clearly written and the newly described fluorescent probe has interesting features. However there is a number of issues with the manuscript that need to be addressed in order to strengthen the message. The application of the probe to cell lines has to be more thoroughly covered and requires additional data. This should the focus of publication as the probe 5 was previously synthesized by the authors as noted by citation of their work (ref 38).

On Page 8 line 245 the authors mention that the place of conversion (azide->amine) remains unclear. Could you elaborate? Looking at Fig 7 there is an overlap between cells location in visible light and the fluorescent signal. Though it is difficult to judge as there is a second layer of the cells in all images. To fully confirm the probe usefulness I would advise to make a control experiment with azide probe (5) incubated in growth medium with FBS. Just to rule out a possibility that conversion of the probe to amine occurs in the growth medium and the amine compound actually enters the cell, giving fluorescent signal in cell. In part the probe 5 entrance to the cells is confirmed by HPLC data (actually there is no reference to supplementary information containing the chromatogram). I also recommend to take an image in confocal microscope after incubation with the probe. Due to high hydrophobicity the probe could localize in the cell membrane. Finally I recommend to perform a simple experiment with cells loaded with azide probe that would undergo a hypoxia, which should provide more intensive fluorescent signal then control cells.

The MS spectra shown on page S8 of the supporting information are zoomed into the region of the expected m/z. Are there any peaks in the region > 400 m/z? I just wonder if any small aggregates (dimers/trimers etc) can be identified on the MS spectra, considering the compound tendency to aggregate.

I suggest to make the title more straightforward and avoid use of “/” sign. Upon completion of the suggested cell studies the title could emphasize the applicability of 4-azidocinnoline as a new probe to detect cellular hypoxia.

Considering that there are ionizable groups is there any dependence of probe 6 fluorescence on pH?

The section about synthesis of additional compounds (7-10) seems disconnected to the rest of the manuscript, which focuses on probe 5 and 6. I would suggest keeping 1-2 sentences about the future possibilities of this synthetic approach and move the scheme to supporting information.

Looking at Fig. 6 I wonder if there is a time dependence of the aggregate formation. At any point were the aggregates visible in the cuvette?

Page 7 lines 220/224 this is a repetition of the text on page 5 lines 160-163.

Figure 7 caption – here I’m confused what does it mean amine 6 suitability as a dye? For what application? The probe 5 ability to detect changes in cells e.g. hypoxia is more clear.

Minor issues:

Page 1, line 22 – please add what the probes would be applicable to?

Page 5 line 161 – please change back-ground to background

Page 6 line 179 – please change enchantment to enhancement

Page 7 line 235 – I believe it is Figure 7B not 4

Page 13 line 472 – Flow cytometry – please add a sentence about the incubation of cells with probe prior to analysis

There is no reference to Fig. 7A

Author Response

See attached files

Round 2

Reviewer 4 Report

The manuscript shows great improvement. The authors' comment to my questions is satisfactory. I agree that since a separate paper, describing biological application of the probe is under preparation there is no need to further expand this here.